# Interactome Profiling of N-Terminus-Truncated NS1 Protein of Influenza A Virus Reveals Role of 14-3-3γ in Virus Replication

**DOI:** 10.3390/pathogens11070733

**Published:** 2022-06-27

**Authors:** Rei-Lin Kuo, Ee-Hong Tam, Chian-Huey Woung, Chu-Mi Hung, Hao-Ping Liu, Helene Minyi Liu, Chih-Ching Wu

**Affiliations:** 1Research Center for Emerging Viral Infections, College of Medicine, Chang Gung University, Taoyuan 33302, Taiwan; rlkuo@mail.cgu.edu.tw (R.-L.K.); e_h2151992@hotmail.com (E.-H.T.); 2Department of Medical Biotechnology and Laboratory Science, College of Medicine, Chang Gung University, Taoyuan 33302, Taiwan; chelsy.woung@gmail.com (C.-H.W.); gimmyhung792@gmail.com (C.-M.H.); 3Graduate Institute of Biomedical Sciences, College of Medicine, Chang Gung University, Taoyuan 33302, Taiwan; 4Division of Allergy, Asthma, and Rheumatology, Department of Pediatrics, Linkou Chang Gung Memorial Hospital, Taoyuan 33305, Taiwan; 5Department of Veterinary Medicine, College of Veterinary Medicine, National Chung Hsing University, Taichung 40227, Taiwan; hpliu@dragon.nchu.edu.tw; 6Graduate Institute of Biochemistry and Molecular Biology, College of Medicine, National Taiwan University, Taipei 10051, Taiwan; mliu@ntu.edu.tw; 7Molecular Medicine Research Center, Chang Gung University, Taoyuan 33302, Taiwan; 8Department of Otolaryngology-Head & Neck Surgery, Linkou Chang Gung Memorial Hospital, Taoyuan 33305, Taiwan

**Keywords:** influenza A virus, N-terminus-truncated NS1, 14-3-3γ, interactome

## Abstract

Influenza A virus is transmitted through a respiratory route and has caused several pandemics throughout history. The NS1 protein of influenza A virus, which consists of an N-terminal RNA-binding domain and a C-terminal effector domain, is considered one of the critical virulence factors during influenza A virus infection because the viral protein can downregulate the antiviral response of the host cell and facilitate viral replication. Our previous study identified an N-terminus-truncated NS1 protein that covers the C-terminus effector domain. To comprehensively explore the role of the truncated NS1 in cells, we conducted immunoprecipitation coupled with LC-MS/MS to identify its interacting cellular proteins. There were 46 cellular proteins identified as the components of the truncated NS1 protein complex. As for our previous results for the identification of the full-length NS1-interacting host proteins, we discovered that the truncated NS1 protein interacts with the γ isoform of the 14-3-3 protein family. In addition, we found that the knockdown of 14-3-3γ in host cells reduced the replication of the influenza A/PR8 wild-type virus but not that of the PR8-NS1/1-98 mutant virus, which lacks most of the effector domain of NS1. This research highlights the role of 14-3-3γ, which interacts with the effector domain of NS1 protein, in influenza A viral replication.

## 1. Introduction

Influenza A virus is a contagious virus that can be spread by respiratory droplets from patients. The virus had caused several pandemics in the 20th century and over hundreds of millions of deaths [1]. The NS1 protein of the influenza A virus is one of the critical virulence factors. The major role of this protein is to downregulate the antiviral response of host cells to facilitate viral replication [2,3]. The NS1 protein has two functional domains, the N-terminal RNA-binding domain (RBD) and the C-terminal effector domain [4].

The RBD of the NS1 protein has been shown to sequester dsRNA from 2′-5′ oligoadenylate synthetase (OAS) in order to inhibit OAS-RNase L activation and, hence, prevent cleavage of viral RNA [5,6]. Several studies have also stated that the RBD competes with RIG-I for dsRNA binding or directly binds to RIG-I to prevent RIG-I activation and inhibit type-I IFN activation [7,8]. Additionally, the RBD can bind to the nucleolar RNA helicase 2 (DDX21) to prevent disruption of viral polymerase assembly through PB1-DDX21 interaction [9]. The effector domain of the NS1 protein also plays a critical role in inhibition of host antiviral response. It has been reported that the effector domain prevents antiviral signal transduction by interacting with inhibitor of nuclear factor kappa-B kinase subunits α and β (IKKα and IKKβ) [10] and suppresses antiviral gene expression and protein production by interacting with human polymerase-associated factor 1 complex (hPAF1C) and poly(A)-binding protein II (PABII) [11,12]. Moreover, the effector domain binds to the cleavage and polyadenylation specificity factor 30 (CPSF30) and globally inhibits maturation of cellular mRNA, including the mRNA of antiviral genes [13,14,15]. Interestingly, not only is the full-length NS1 protein expressed in influenza A virus-infected cells, N-terminus-truncated NS1 (tNS1) proteins can also be produced during viral infection. We previously identified two tNS1 proteins of the A/PR8/H1N1 virus, translated from the second and third in-frame AUGs (codons 79 and 81, respectively) of the NS transcript. In addition, elimination of the tNS1 proteins increases IRF3 phosphorylation and IFN-β transcription and leads to reduction of viral replication, suggesting that the tNS1 proteins play a role in inhibition of the type-I IFN signaling pathway [16].

Interactome analysis with a proteomic approach is a practical tool to identify protein functions. To comprehensively study the cellular biological processes regulated by influenza A NS1, we previously analyzed the NS1 interactome using liquid chromatography-tandem mass spectrometry (LC-MS/MS) and found that the NS1 protein can interact with host factors involved in RNA splicing/processing, cellular macromolecule localization, protein transport/targeting, and protein folding [17]. We further demonstrated that pre-mRNA-processing factor 19 (PRP19) is a positive regulator for replication of influenza A virus, whereas 14-3-3ε serves as a negative regulator during viral infection [17,18]. To date, seven 14-3-3 isoforms have been identified in mammalian cells, including 14-3-3β, ε, γ, η, σ, θ, and ζ isoforms. The 14-3-3 family proteins are highly conserved and expressed in all eukaryotic cells. It has been reported that these proteins can interact with hundreds of proteins and regulate a wide range of biological processes, such as cell cycle regulation, protein trafficking, and apoptotic and autophagy signaling [19,20,21,22,23,24,25]. Our previous study demonstrated that 14-3-3ε interacts with the N-terminal domain of NS1, and that knockdown of 14-3-3ε leads to reduction of IRF3 phosphorylation and IFN-β transcription during influenza A virus infection and increases the viral replication [18].

Our previous results showed that the tNS1 protein has an inhibitory role in regulating type-I IFN response. The tNS1 proteins mainly localize in the cytoplasm, whereas the full-length NS1 protein resides in the nucleus [16], suggesting that it is necessary to further understand the role of the tNS1 proteins in host cells through identification of the host proteins that interact with the tNS1 proteins. We thus applied a plasmid that expresses amino acids 79–230 of NS1, covering the effector domain of NS1 and denoted tNS1_79-230_, to further investigate the role of tNS1 in regulation of cellular processes and viral replication. In the present study, we identified 14-3-3γ as a tNS1_79-230_-interacting partner by using co-immunoprecipitation and LC-MS/MS approaches. Our results further demonstrated that knockdown of 14-3-3γ expression reduces influenza A virus replication. In addition, replication of a mutant influenza A virus lacking an intact NS1 effector domain is not affected by 14-3-3γ. Overall, this research reveals a positive role for 14-3-3γ in influenza A virus replication.

## 2. Results

### 2.1. Interactome Analysis of N-Terminus-Truncated NS1 Proteins of Influenza A Virus

NS1 proteins of influenza A virus play vital roles in the regulation of antiviral responses in host cells. To comprehensively explore the functions of NS1 protein, we previously identified host proteins that interact with the NS1 protein [17]. Two N-terminus-truncated NS1 (tNS1) proteins, which cover the effector domain of the full-length NS1 protein, have been found to be expressed in cells infected with influenza A virus and involved in regulation of viral replication [16]. Noteworthily, most of the tNS1 protein is located in the cytoplasm, whereas the full-length NS1 protein is mainly localized in the nucleus [16]. The host proteins that interact with the tNS1 proteins therefore require further investigation to reveal the regulatory role of the tNS1 proteins in host cells. To determine protein components in the tNS1-associated protein complex, 293T cells were transfected with a control vector and a plasmid expressing FLAG-tagged tNS1_79-230_ encoded by the influenza A virus PR8(H1N1) strain. After 24 h, lysates of the transfected cells were subjected to immunoprecipitation (IP) with anti-FLAG resin (Figure 1A and Appendix A). The IP products of the control and the tNS1_79-230_ groups were then detected with a GeLC-MS/MS method (Appendix A). The analysis led to identification of 1338 non-redundant proteins with at least two unique peptides. Among these, 955 (71.38%) were detected in both groups, whereas 116 (8.67%) and 267 (19.96%) were exclusively found in the control and tNS1_79-230_ groups, respectively (Figure 1B and Appendix A).

To determine the reproducibility of the mass spectrometry analyses, overlap analyses for the proteins identified in each replicate were further performed. In total, 1071 (80.04%) and 1222 (91.33%) proteins were found in the triplicates of the control and tNS1_79-230_ groups, respectively (Figure 1B,C). At least 80% of the proteins could be detected in either the duplicates or the triplicates of the experiments, whereas around 19% were present in single replicates (Figure 1C and Appendix A). Moreover, the false discovery rate (FDR) of peptide identification was ascertained using a decoy database. All FDRs in each replicate were less than 0.05%. The results collectively suggest that the protein identification was becomingly performed.

### 2.2. Spectral Counting-Based Identification of Proteins Interacting with tNS1_79-230_ Protein

To identify candidate proteins interacting with the tNS1_79-230_ proteins, we applied the spectral counting-based approach to quantify the relative amounts of proteins in the immunoprecipitants (Appendix A). To calculate the ratios of each protein, the average spectral count (SC) of the protein in the tNS1_79-230_ group was divided by that in the control group. The mean (1.650) and SD (1.968) of the ratios of total proteins were then obtained. Proteins with a ratio (5.586) more than two SDs above the mean ratio and observed in at least two replicates of the tNS1_79-230_ group were considered proteins interacting with tNS1 protein (Appendix A). As shown in Table 1, 46 proteins were determined based on these cutoffs. Among these, 43 were uniquely present in the tNS1_79-230_ group, and the levels of three proteins in the tNS1_79-230_ group were higher than those in the control group (Table 1). The proteins with ratios between the mean plus one and two SDs (3.618–5.586) are additionally listed in Appendix A to demonstrate that the most of the potential interacting partners of tNS1 protein were included in the present analysis.

### 2.3. Determination of Biological Pathways Involved with the tNS1-Interacting Partners

To understand the biological meaning behind the existence of tNS1-associated protein complexes, the 46 proteins (Table 1) were functionally annotated with the online software DAVID. As shown in Table 2, biological processes, including cornification/keratinization, neutrophil degranulation, regulation of protein kinase activity, and translation/rRNA processing, were enriched. Furthermore, biological pathway analysis using the Reactome pathway database demonstrated that the tNS1-interacting proteins are likely involved in keratinization, rRNA processing/eukaryotic translation, neutrophil degranulation, programmed cell death, and Rho GTPase signaling pathways (Table 3).

We further used the STRING online database to establish a network of protein–protein interactions (PPIs) between the 46 proteins, and 80 strong interaction links between individual nodes/proteins were depicted in the PPI network (Figure 2). In line with the results of the enrichment analyses using DAVID and the Reactome Pathway Database (Table 2 and Table 3), the PPI networks correlated with cornification/keratinization, translation/rRNA processing, and neutrophil degranulation were identified (Figure 2). Noteworthily, one module was revealed in the STRING analysis that depicted the interactions of 14-3-3γ with proteins associated with Rho GTPase and programmed cell death signaling pathways (Figure 2).

### 2.4. N-Terminus-Truncated NS1 of PR8 Virus Interacts with 14-3-3γ Protein

It has been found that two 14-3-3 isoforms, ε and η, are involved in translocating the viral RNA sensors RIG-I and MDA5, respectively, and therefore contribute to activation of type-I IFN expression in RNA virus-infected cells [26,27,28]. In our previous study, we also demonstrated that 14-3-3ε, which interacted with the RBD of influenza A NS1 protein, acts as an antiviral protein during influenza A virus infection [18]. Since the interactome analysis revealed several isoforms in the 14-3-3 protein family to be tNS1-interacting partners, we performed immunoprecipitation to verify the interaction. Among the isoforms, we initially examined the interaction of NS1 and 14-3-3γ in 293T cells co-transfected with Myc-tagged 14-3-3γ and either 3✕FLAG-tagged NS1 or tNS1_79-230_-expressing plasmids. The result showed that the tNS1_79-230_ protein associated with 14-3-3γ more strongly than the full-length NS1 did (Figure 1A). With this approach, we confirmed that the effector domain of the NS1 protein may be able to interact with 14-3-3γ.

### 2.5. Knockdown of 14-3-3γ Expression Reduces Replication of Influenza A PR8 Virus

Since we identified 14-3-3γ as an NS1-binding partner, we further investigated the role of 14-3-3γ in influenza A virus replication. To this end, replication of influenza A PR8 wild-type virus was determined with A549 cells in which 14-3-3γ expression was inhibited using siRNA specific to 14-3-3γ. We discovered that 14-3-3γ knockdown resulted in reduced replication of the virus (Figure 3A). As our data showed that 14-3-3γ could interact with the effector domain of the NS1 protein (Figure 1A), the role of 14-3-3γ in viral replication was also examined using the PR8-NS1/1-98 mutant virus, which lacks most (amino acids 99–230) of the NS1 effector domain. As shown in Figure 3B, knockdown of 14-3-3γ expression did not affect the replication of the PR8-NS1/1-98 mutant virus. Collectively, the results strongly suggest that 14-3-3γ acts as a positive regulator for influenza A virus replication through the interaction with the viral NS1 protein.

It has been found that two 14-3-3 isoforms, ε and η, are involved in activation of type-I IFN expression, which serves as the first line of antiviral response [26,28]. We further determined whether 14-3-3γ is involved in regulation of type-I IFN expression and impacts influenza A virus replication. 14-3-3γ expression of A549 cells was knocked down and then the cells were infected with PR8 wild-type virus at an MOI of 2. At 3, 6, 9, and 12 h post-infection, total RNA and cell lysates were collected to detect IFN-β mRNA and phosphorylation of IRF3, respectively. As shown in Figure 4, reduction of 14-3-3γ expression did not change the levels of IFN-β mRNA and IRF3 phosphorylation. Therefore, the reduced replication of PR8 wild-type virus in the 14-3-3γ knockdown cells might not be caused by the type-I IFN response.

### 2.6. Knockdown of 14-3-3γ Did Not Affect Influenza A Virus RNA Polymerase Activity

The observations described above showed that 14-3-3γ could serve as a supportive role in viral replication. We further determined whether knockdown of 14-3-3γ expression dysregulates viral RNA synthesis during infection. A549 cells were transfected with 14-3-3γ siRNA and then infected with PR8 wild-type virus at an MOI of 0.001 for 36 h. Three species of viral RNAs—viral genome RNA (vRNA), complementary RNA (cRNA), and viral messenger RNA (mRNA)—in the infected cells were quantified with RT-qPCR. As shown in Figure 5, knockdown of 14-3-3γ significantly decreased the levels of vRNA, cRNA, and viral mRNA in the cells infected with PR8 virus.

Additionally, since previous reports showed that NS1 can upregulate viral RNA synthesis by enhancing viral polymerase activity [29,30], we investigated whether the reducing replication of PR8 wild-type virus under 14-3-3γ knockdown was caused by a change in NS1-mediated enhancement of viral polymerase activity. To this end, the minigenome assay was applied to examine the viral polymerase activity. 14-3-3γ expression of 293T cells was first knocked down and then the cells were cotransfected with the plasmids encoding PA, PB1, PB2, NP, and NS1 proteins and a plasmid-expressing virus-like RNA containing the antisense sequence of the firefly luciferase open reading frame. The luciferase activity, which represents the viral polymerase activity, was measured at 24 h post-transfection. The results showed that the viral polymerase activities could be detected along with the essential viral polymerase complex and NP (Figure 6A, lanes 5 and 6). Moreover, we found that coexpression of NS1 significantly increased luciferase activities, which is consistent with previous findings (Figure 6A, lanes 7 and 8). However, knockdown of 14-3-3γ expression did not affect the viral polymerase activity with or without NS1 coexpression (Figure 6A, lane 5 vs. 6 and lane 7 vs. 8). We also examined whether the tNS1_79-230_ protein could be involved in viral polymerase activity. We found that overexpression of tNS1_79-230_ did not enhance the viral polymerase activity (Figure 6B, lane 3 vs. 5). Consistently, knockdown of 14-3-3γ expression had no effect on viral polymerase activity in cells overexpressing tNS1_79-230_ (Figure 6B, lane 4 vs. 6). These results demonstrated that 14-3-3γ may not be involved in regulation of viral polymerase activity, suggesting that 14-3-3γ might regulate viral replication by regulating other stages of the viral life cycle.

## 3. Discussion

Our previous study showed that the tNS1 protein, which contains the effector domain of the NS1 protein, encoded by influenza A PR8 virus contributes to suppression of IRF3 activation. The PR8 virus-eliminating tNS1 protein increases IRF3 phosphorylation and IFN-β mRNA expression in infected cells and leads to decreasing viral replication [16]. The results demonstrate that the effector domain of NS1 plays a role in downregulating type-I IFN expression. To comprehensively explore the roles of tNS1 in cells, we conducted immunoprecipitation coupled with LC-MS/MS to identify tNS1-interacting cellular proteins. We discovered several isoforms of the 14-3-3 protein family in the tNS1 protein complex. We further confirmed the interaction between 14-3-3γ and either full-length NS1 or tNS1 through immunoprecipitation. In addition, we found that the inhibition of 14-3-3γ expression in the host cells reduced the replication of the PR8 wild-type virus, but not that of the PR8-NS1/1-98 mutant virus, which lacks most of the effector domain of NS1. This result suggests a correlation between the interaction of 14-3-3γ and NS1 and viral replication. Since, upon PR8 wild-type virus infection, neither the viral polymerase activity nor the IFN-β expression are affected by the decreasing level of 14-3-3γ, the detailed mechanism for the involvement of 14-3-3γ in influenza A virus replication requires further elucidation.

There were 46 cellular proteins identified as the interacting partners of tNS1 (Table 1). In contrast to a previous analysis of full-length NS1 interactome, which showed that the full-length NS1 of the PR8 virus interacts with 64 host proteins [17], only 10 cellular proteins (FLG2, IMPDH2, CCT2, MDH2, DDX47, ECHDC1, PNO1, YWHAG, NSA2, and DDX39B) were here detected in both the full-length and truncated NS1-associated protein complexes. There were 54 and 36 proteins that exclusively interacted with full-length and truncated NS1 proteins, respectively (Appendix A). Pathway-wise analysis has demonstrated that the proteins interacting with full-length NS1 are involved in the cellular processes of RNA splicing/processing, the cell cycle, and protein folding/targeting [17]. However, the components in tNS1 protein complexes (Table 1) are mostly involved in cornification, neutrophil degranulation, rRNA processing/eukaryotic translation, programmed cell death, and Rho GTPase signaling (Table 2 and Table 3), suggesting that tNS1 plays a regulatory role in viral infection via a mechanism that is different from that of full-length NS1.

The human 14-3-3 proteins comprise seven isoforms (β, ε, γ, η, σ, θ, and ζ). They are highly conserved in the amino acid sequences and share a similar structure [31,32,33]. Due to the high similarity in the sequence and structure, loss of function in a particular 14-3-3 protein can be compensated by others [34,35,36,37], suggesting that the functions of specific 14-3-3 isoforms may be contained in different isoforms. Previously, several studies demonstrated the role of 14-3-3 proteins in activation of type-I IFN [26,28]. Our previous study also showed that the full-length NS1 of the PR8 virus interacts with 14-3-3ε, and that the interaction may contribute to suppression of type-I IFN expression. In the current study, we investigated the role of 14-3-3γ in regulation of IFN-β activation. However, the data demonstrated that reducing 14-3-3γ expression results in no change in IFN-β expression, though it actually slows PR8 wild-type virus replication.

In addition to inhibition of host innate immune response, NS1 has also been shown to regulate viral RNA synthesis by enhancing viral polymerase activity [29,30,38]. Since our results demonstrated that 14-3-3γ plays a positive role in viral replication of the PR8 virus, we further investigated whether 14-3-3γ facilitates NS1-mediated enhancement of viral polymerase activity. Consistent with the previous findings, our results for the minigenome polymerase activity assay showed that overexpression of NS1 enhances influenza A polymerase activity. However, no alteration in viral polymerase activity was detected in 14-3-3γ knockdown cells. These results demonstrated that the supportive role of 14-3-3γ in influenza A virus replication may not be related to the regulation of viral polymerase activity. Therefore, 14-3-3γ may regulate viral replication through the involvement of other steps of the viral life cycle, such as entry, assembly, and budding.

In summary, we identified 14-3-3γ as one of the tNS1-interacting host proteins using an LC-MS/MS approach and revealed that 14-3-3γ plays a proviral role during influenza A virus infection. Despite the detailed mechanisms remaining unclear, our results provide a new insight into the role of 14-3-3γ in viral infection, and these findings will be the cornerstone of future research.

## 4. Materials and Methods

### 4.1. Cells, Plasmids, and Viruses

Human Embryonic Kidney 293T cell line (293T), adenocarcinomic human alveolar basal epithelial cell line (A549), and Madin Darby Canine Kidney (MDCK) cell line (obtained from the American Type Culture Collection (ATCC), Rockville, MD, USA) were cultivated in DMEM (Gibco, Waltham, MA, USA) supplemented with 10% fetal bovine serum (Gibco), 1% non-essential amino acids (Gibco), 1% penicillin-streptomycin (Gibco), and 1% L-glutamine (Gibco). Cells were maintained and handled as described previously [18]. The cDNA of the full-length amino acids and amino acids 79–230 of NS1 from influenza virus A/PR8/34 (H1N1) were cloned into pcDNA3 or p3×FLAG-Myc-CMV26 vectors (Sigma-Aldrich, St. Louis, MO, USA) to generate untagged NS1, 3×FLAG-NS1, and 3×FLAG-tNS1_79-230_ expression plasmids. The Myc-tagged 14-3-3γ-expressing plasmid was provided by Dr. Helene M. Liu from National Taiwan University. Plasmids expressing influenza A PR8/H1N1 viral polymerase subunits (PB1, PB2, and PA) and NP were constructed in the pcDNA3 vector. The luciferase reporter plasmids for determining viral polymerase activity were designated as in previous research [39]. Viruses used in this study were generated by a plasmid-based reverse genetics system, as described previously [18]. The rescued viruses were propagated in 10-day-old embryonated eggs, and the virus titers were determined with a plaque formation assay with monolayers of MDCK cells.

### 4.2. Immunoprecipitation, Immunoblotting, and Antibodies

293T cells were cotransfected with Myc-tagged 14-3-3γ and either 3×FLAG-NS1 or 3×FLAG-tNS1_79-230_ using TransIT-LT1 transfection reagent (Mirus, Madison, WI, USA). At 48 h post-transfection, the cells were lysed with a buffer containing 100 mM Tris-HCl (pH 7.5), 250 mM NaCl, 0.5% sodium deoxycholate, 1 mM phenylmethylsulfonyl fluoride (PMSF), and 0.5% NP-40. The lysates were then subjected to immunoprecipitation using anti-FLAG M2 resin (Cat. No. A2220, Sigma-Aldrich) and eluted with 3 × FLAG peptide (Cat. No. F4799, Sigma-Aldrich). Cell lysates and immunoprecipitated products were separated with SDS-PAGE and transferred to PVDF membranes. After blocking with 5% skim milk, the membranes were incubated separately with primary antibodies and reacted species-specific secondary antibodies after washing and then detected with an Immobilon Western Chemiluminescent HRP Substrate (Millipore, Bedford, MA, USA). The following primary antibodies were used in the research: anti-Myc antibody (Cat. No. M4439, Sigma-Aldrich), anti-FLAG-M2 (Cat. No. F1804, Sigma-Aldrich), anti-14-3-3γ (Cat. No. 5522S, Cell Signaling, Danvers, MA, USA), anti-phosphorylated IRF3 (Cat. No. 4947; Cell Signaling), anti-total IRF3 (Cat. No. sc-9082; Santa Cruz Biotechnology, Santa Cruz, CA, USA), anti-β-tubulin (Cat. No. LT9991, LifeTein, NJ, USA; Cat. No. MA5-16308, Thermo Fisher Scientific, Waltham, MA, USA), and anti-NP and anti-NS1 (both were kindly provided by Dr. Shin-Ru Shih).

### 4.3. Reverse-Phase LC-MS/MS Analysis and Protein Identification

The immunoprecipitates of tNS1_79-230_ were segregated with SDS-PAGE and stained with a Colloidal Blue Staining kit (Thermo Fisher Scientific). The gel lanes were sliced into twenty fractions, and each fraction was equally divided into three pieces to provide technical replicates. After in-gel digestion with trypsin, the peptides were extracted and analyzed with reversed-phase LC-MS/MS as previous described [40]. The Mascot algorithm (version 2.1, Matrix Science, Boston, MA, USA) was used to search for the acquired MS/MS spectra against the Swiss-Prot human sequence database (released 02 March 2022, selected for Homo sapiens, 20,376 entries) of the European Bioinformatics Institute. The search results were further integrated using Scaffold software (version 3.6.5; Proteome Software, Portland, OR, USA). The criteria for the protein database search and the process for the search files in Scaffold software have been previously described [40]. Probabilities of PeptideProphet and ProteinProphet ≥ 0.95 were applied to ensure that the false discovery rate of protein identification was lower than 0.5%. Only proteins with at least two unique peptides were retained in this study.

### 4.4. Bioinformatics Analysis

To identify components in the tNS1_79-230_ protein complex, we compared the protein levels between the immunoprecipitation products of the control and tNS1_79-230_ vector groups with a spectral counting-based quantification approach, as previously described [40]. Briefly, the spectral numbers of the identified proteins were first exported to Microsoft Excel format with Scaffold software. The spectral count (SC) of each protein was then divided by the SC of total proteins to obtain the normalized SC of each protein. The fold change of proteins between the control and tNS1_79-230_ groups was estimated as the ratio of the average of normalized SCs in the tNS1_79-230_ group to that in the control group. Due to the fact that not all proteins were detected in each replicate, the SCs for unidentified proteins or missing values in a certain replicate were assigned the number one to avoid overestimating the fold changes and dividing by zero.

The biological processes and pathways involved in the tNS1_79-230_-interacting partners were revealed by the Database for Annotation, Visualization, and Integrated Discovery (DAVID, 2021 Update, https://david.ncifcrf.gov/, accessed on 8 March 2022) and the Reactome pathway database, respectively [40]. The known and predicted associations between the tNS1_79-230_-inteacting proteins were analyzed with STRING online software (version 11.5, https://string-db.org/, accessed on 8 March 2022). A combined score of confidence ≥ 0.7 was used as the cutoff criterion [40].

### 4.5. Reverse Transcription and Quantitative PCR

Total RNA from the mock or infected cells was prepared using TRIzol reagent (Invitrogen, Carlsbad, CA, USA). The reverse transcription (RT) and quantitative PCR (qPCR) were performed as previously described [18]. The specific primers used in this research are listed below: RT primers for 18S rRNA: 5′-CCATCCAATCGGTAGTAGCG-3′, influenza virus NP vRNA: 5′-AGCAAAAGCAGGGT-3′, and influenza virus NP cRNA: 5′-AGTAGAAAACAAGGGTA-3′; qPCR primers for 18S rRNA: 5′-GTAACCCGTTGAACCCCATT-3′ (forward) and 5′-CCATCCAATCGGTAGTAGCG-3′ (reverse), IFN-β: 5′-CAGTCTGCACCTGAAAAGATATTATG-3′ (forward) and 5′-GATTTCCACTCTGACTATGGTCCAGG-3′ (reverse), and influenza virus NP: 5′-GAGCTCTCGGACGAAA-3′ (forward) and 5′-CCTCTGCATTGTCTCC-3′(reverse).

### 4.6. RNA Interference

A549 cells were transfected with 60 nM 14-3-3γ siRNA or universal control siRNA using Lipofectamine 3000 (Invitrogen). The siRNA sequences used for 14-3-3γ were 5′-GAGGAAUGGCCCUCAUUCA-3′ and 5′-UGAAUGAGGGCCAUUCCUC-3′, and those for the universal control were 5′-GAUCAUACGUGCGAUCAGA-3′ and 5′-UCUGAUCGCACGUAUGAUC-3′. At 24 h post-transfection, cells were infected with PR8 wild-type virus or PR8-NS1/1-98 virus at an MOI of 0.001 for the indicated time periods. The titers of the virus in the supernatants and the relative amounts of viral RNAs in the infected cells were determined with a plaque formation assay and RT-qPCR, respectively, to evaluate the effect of 14-3-3γ knockdown on viral replication. To assess the role of 14-3-3γ in type-I IFN activation, A549 cells transfected with 14-3-3γ siRNA for 24 h were infected with PR8 wild-type virus at an MOI of 2. At 9 h post-infection, the total cellular RNA was extracted for IFN-β mRNA detection, and the cell lysates were collected for immunoblotting.

### 4.7. Minigenome Polymerase Activity Assay

293T cells transfected with 14-3-3γ siRNA for 24 h were further cotransfected with a vRNA-like firefly luciferase reporter plasmid [39], a renilla luciferase control plasmid, and plasmids expressing influenza A/PR8 PA, PB1, PB2, NP, and NS1, respectively. At 24 h post-transfection, the cell lysates were harvested and the luciferase activity was detected with a dual-luciferase reporter assay (Cat. No. E1910, Promega, Madison, WI, USA) following the manufacturer’s instructions.

### 4.8. Statistical Analysis

The minigenome assay was analyzed using one-way ANOVA with Tukey’s multiple-comparison test (GraphPad Prism version 8.0.0, La Jolla, CA, USA). The comparisons of host mRNA, viral RNAs, and viral titers between two groups were analyzed using unpaired Student’s *t*-tests. Data are presented as means ± standard deviation (SD). Results with *p* values less than 0.05 were considered statistically significant. *, *p* < 0.05; **, *p* < 0.01; ***, *p* < 0.001; ****, *p* < 0.0001.

## Figures and Tables

**Figure 1 pathogens-11-00733-f001:**
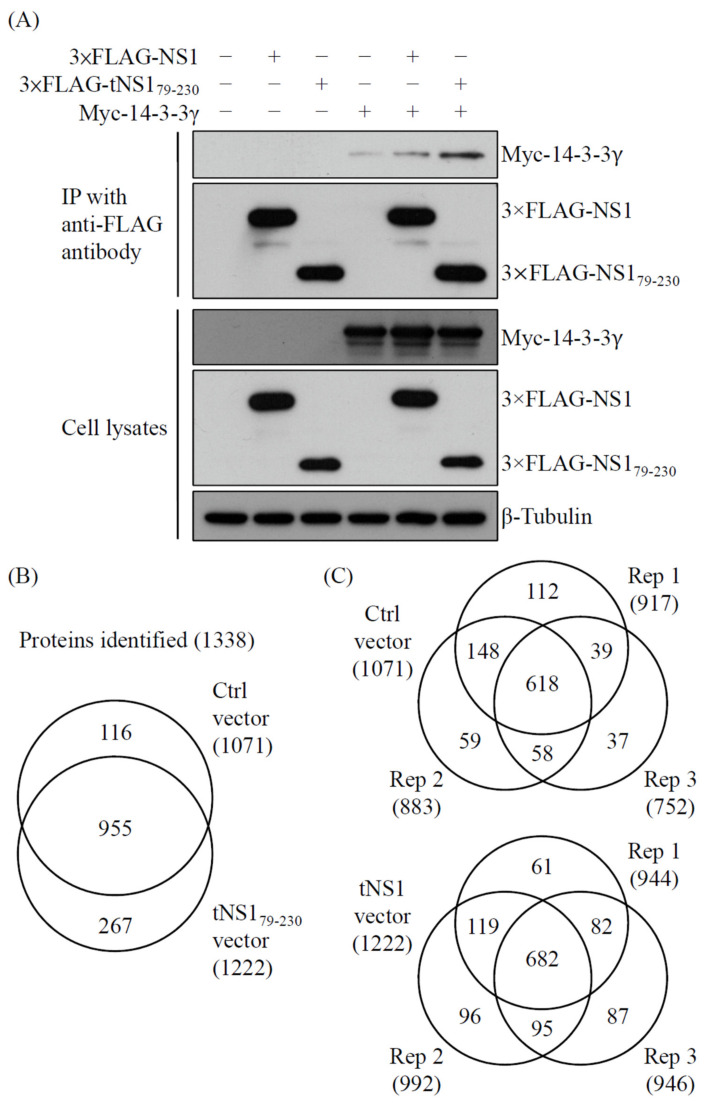
Validation of the interaction between 14-3-3γ and influenza A viral NS1 proteins. 293T cells were cotransfected with vectors of control, 3×FLAG-NS1, 3×FLAG-tNS1_79-230_, and/or Myc-tagged 14-3-3γ. At 48 h post-transfection, cell lysates were collected and subjected to anti-FLAG immunoprecipitation. (**A**) The precipitates and lysates were analyzed by immunoblotting with anti-Myc, anti-FLAG, and anti-β-tubulin antibodies, respectively. (**B**) The precipitated proteins were separated with SDS-PAGE, stained with a Colloidal Blue Staining kit, and detected with LC-MS/MS. Venn diagrams show overlaps between the proteins identified in the control and the tNS1_79-230_ groups. (**C**) Venn diagrams display overlaps between the proteins identified in the three replicates. The total numbers of identified proteins are listed in brackets.

**Figure 2 pathogens-11-00733-f002:**
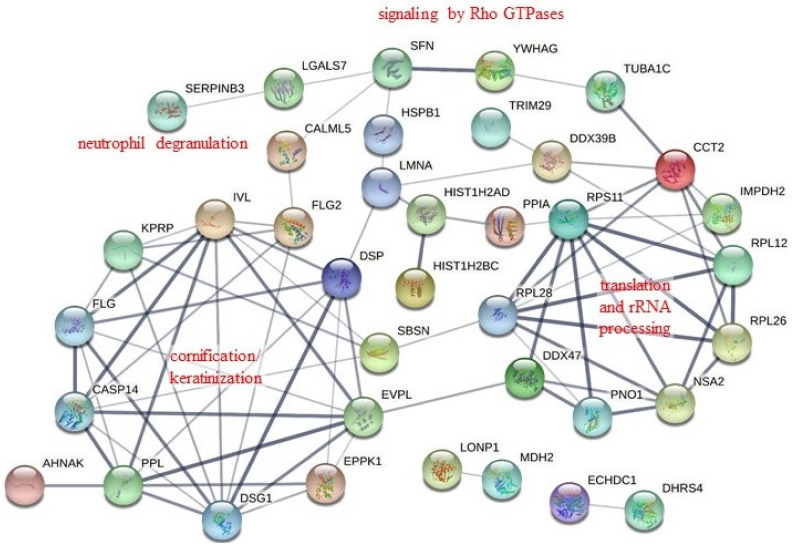
Protein–protein interaction (PPI) network for the proteins co-immunoprecipitating with the H1N1 N-terminus-truncated NS1 protein. The PPI network was constructed with the 46 proteins listed in Table 1 using the online STRING database (v11.5, http://string-db.org/, accessed on 8 March 2022). The network depicts 80 interaction links between individual nodes/proteins (solid lines). One module was identified in the STRING analysis that depicted the interactions of 14-3-3γ with proteins involved in Rho GTPase and programmed cell death signaling pathways.

**Figure 3 pathogens-11-00733-f003:**
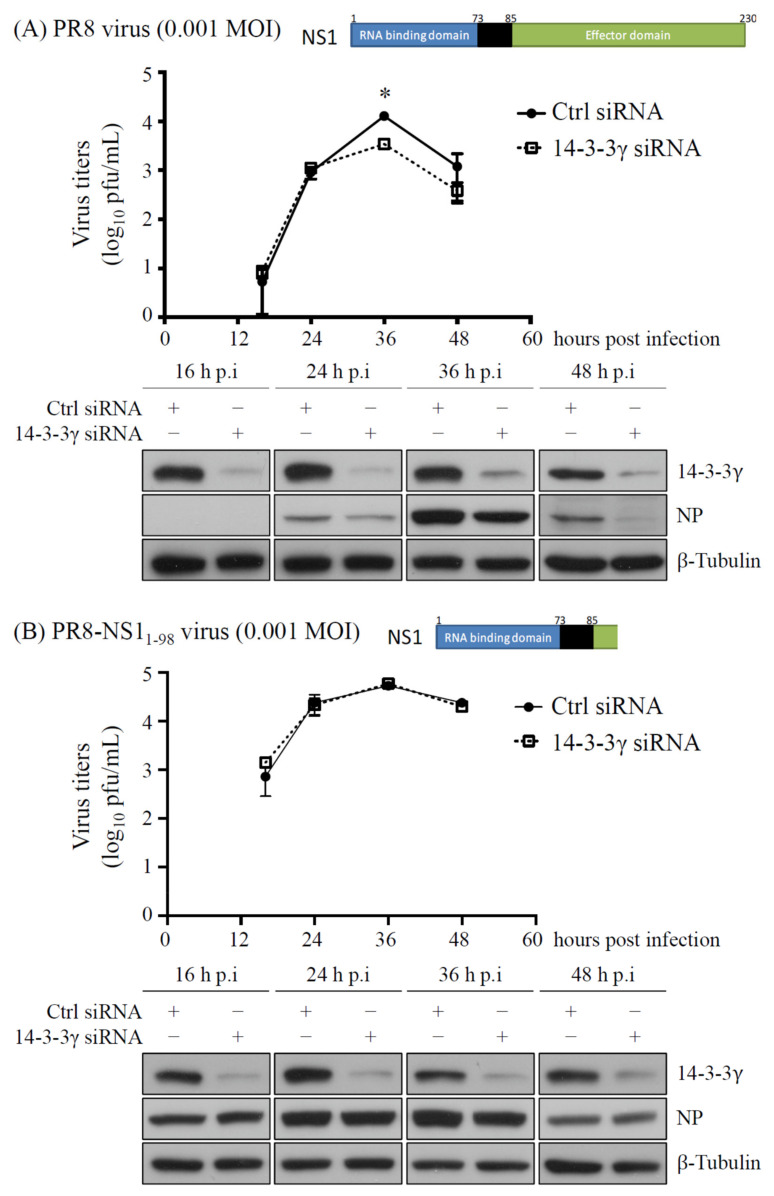
Determination of influenza A virus replication in cells with knocked-down 14-3-3γ expression. A549 cells were transfected with 14-3-3γ siRNA for 24 h and then infected with either influenza A wild-type virus PR8 (**A**) or PR8-NS1/1-98 mutant virus (**B**) at an MOI of 0.001. At the indicated time points, the supernatants of the infected cells were collected for titration of the infectious viral particles with a plaque formation assay. The cells were extracted to analyze the protein expression by immunoblotting with anti-14-3-3γ, anti-influenza A virus NP, and anti-β-tubulin antibodies. The experiments were performed in triplicate and the results were analyzed with Student’s *t*-test. *, *p* < 0.05.

**Figure 4 pathogens-11-00733-f004:**
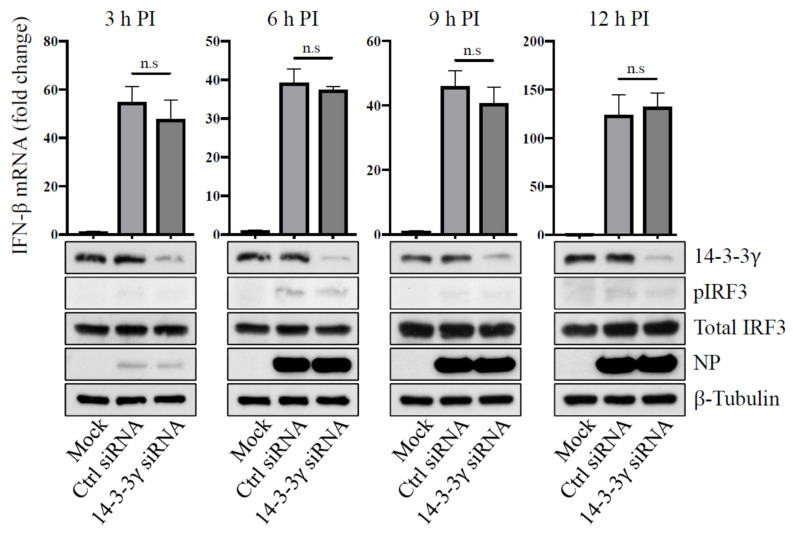
Examination of IFN-β mRNA expression in 14-3-3γ knockdown cells infected with influenza A virus. A549 cells were transfected with 14-3-3γ siRNA for 24 h and then infected with influenza A PR8 wild-type virus at an MOI of 2. At 3, 6, 9, and 12 h post-infection (PI), the total RNA of the infected cells was extracted to determine IFN-β mRNA expression with RT-qPCR. Cell extracts were analyzed for 14-3-3γ, phosphorylated IRF3 (pIRF3), total IRF3, influenza virus NP, and β-tubulin with immunoblotting.

**Figure 5 pathogens-11-00733-f005:**
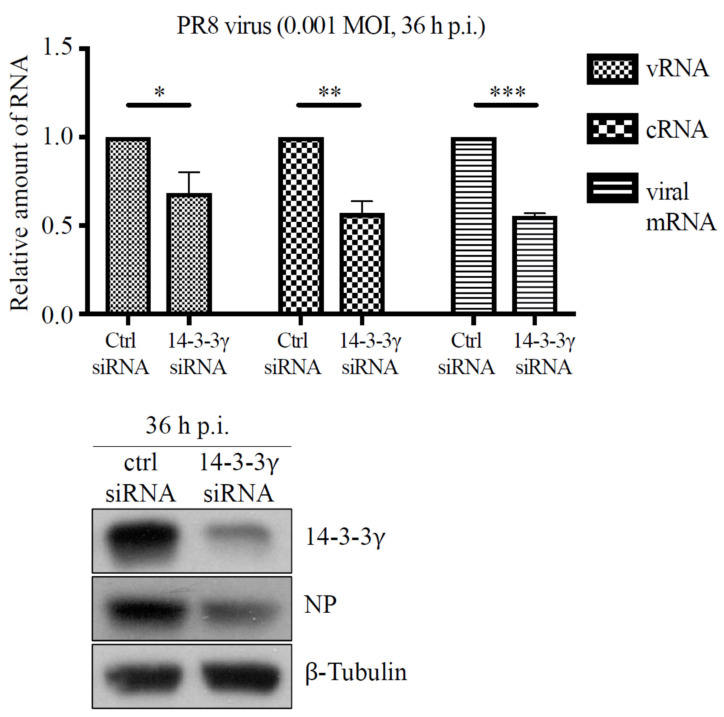
Analysis of influenza A viral RNAs in 14-3-3γ knockdown cells. 14-3-3γ expression of A549 cells was knocked down and then the cells were infected with influenza A PR8 wild-type virus for 36 h. Total RNA of the infected cells was extracted to examine the levels of vRNA, cRNA, and viral mRNA, respectively, using RT-qPCR (right panel). The protein expression of 14-3-3γ, influenza A NP, and β-tubulin was assayed with immunoblotting (left panel). The experiments were performed in triplicate and the results were analyzed with Student’s *t*-test. *, *p* < 0.05; **, *p* < 0.01; ***, *p* < 0.001.

**Figure 6 pathogens-11-00733-f006:**
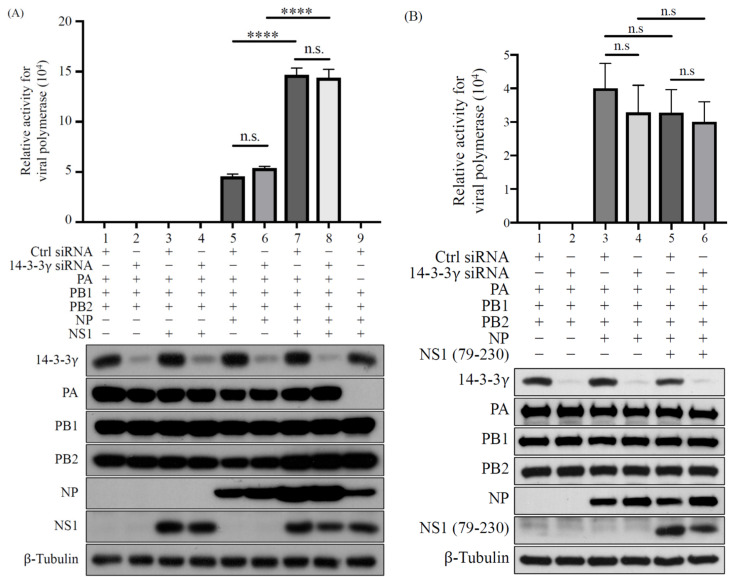
Determination of the regulatory role of 14-3-3γ in the activity of influenza A polymerase. 293T cells were transfected with either control or 14-3-3γ siRNA for 24 h, and then further cotransfected with plasmids expressing influenza A polymerase complex, NP, and NS1 proteins and a vRNA-like luciferase reporter plasmid (**A**) or cotransfected with a plasmid-expressing tNS1_79-230_ protein instead of the full-length NS1-expressing plasmid (**B**). The luciferase activity was measured at 24 h after plasmid transfection. The relative luciferase activity was analyzed and compared using one-way ANOVA with Tukey’s multiple-comparison test. n.s. indicates non-significant. ****, *p* < 0.0001.

**Table 1 pathogens-11-00733-t001:** Spectral counting-based identification of proteins co-immunoprecipitating with H1N1 N-terminus-truncated NS1 protein.

Protein Name (Accession Number, Gene Name)	Spectral Counts (SCs) in Replicates 1/2/3		tNS1/VC Ratio ^a^
	Vector Control (VC)	N-Terminus-Truncated NS1 (tNS1)	
Filaggrin (FILA_HUMAN, FLG)	0/0/0	46/32/0	28.634
Filaggrin-2 (FILA2_HUMAN, FLG2)	0/0/0	34/27/0	22.367
Prelamin-A/C (LMNA_HUMAN, LMNA)	0/0/0	22/24/0	16.767
Calmodulin-like protein 5 (CALL5_HUMAN, CALML5)	0/0/0	21/24/0	16.384
Inosine-5′-monophosphate dehydrogenase 2 (IMDH2_HUMAN, IMPDH2)	0/0/0	18/13/11	15.678
14-3-3 protein gamma (1433G_HUMAN, YWHAG)	0/0/0	13/14/14	15.307
Desmoglein-1 (DSG1_HUMAN, DSG1)	0/0/0	18/22/0	14.567
Galectin-7 (LEG7_HUMAN, LGALS7)	0/0/0	21/18/0	14.384
Keratinocyte proline-rich protein (KPRP_HUMAN, KPRP)	0/0/0	8/9/17	12.935
14-3-3 protein sigma (1433S_HUMAN, SFN)	0/0/0	15/20/0	12.750
Neuroblast differentiation-associated protein AHNAK (AHNK_HUMAN, AHNAK)	0/0/0	13/20/0	11.983
Involucrin (INVO_HUMAN, IVL)	0/0/0	15/16/0	11.417
Envoplakin (EVPL_HUMAN, EVPL)	0/0/0	13/17/0	10.983
Periplakin (PEPL_HUMAN, PPL)	0/0/0	12/17/0	10.600
Histone H2B type 1-C/E/F/G/I (H2B1C_HUMAN, HIST1H2BC)	0/0/0	12/12/4	10.216
Histone H2A type 1-D (H2A1D_HUMAN, HIST1H2AD)	0/0/0	13/11/0	8.983
60S ribosomal protein L26 (RL26_HUMAN, RPL26)	0/0/0	5/9/10	8.957
Suprabasin (SBSN_HUMAN, SBSN)	0/0/0	12/12/0	8.933
60S ribosomal protein L12 (RL12_HUMAN, RPL12)	0/0/0	10/8/6	8.924
Desmoplakin (DESP_HUMAN, DSP)	20/20/0	159/160/23	8.532
40S ribosomal protein S11 (RS11_HUMAN, RPS11)	3/0/0	12/11/14	8.166
Serine/threonine-protein kinase 38 (STK38_HUMAN, STK38)	0/0/0	9/7/5	7.804
Ethylmalonyl-CoA decarboxylase (ECHD1_HUMAN, ECHDC1)	0/0/0	6/8/7	7.795
Heat shock protein beta-1 (HSPB1_HUMAN, HSPB1)	4/2/0	28/18/5	7.684
Spliceosome RNA helicase DDX39B (DX39B_HUMAN, DDX39B)	0/0/0	9/5/6	7.541
Unconventional myosin-Ib (MYO1B_HUMAN, MYO1B)	0/0/0	7/12/2	7.491
Hydroxyacylglutathione hydrolase, mitochondrial (GLO2_HUMAN, HAGH)	0/0/0	5/7/7	7.078
Tubulin alpha-1C chain (TBA1C_HUMAN, TUBA1C)	0/0/0	10/8/0	6.833
Epiplakin (EPIPL_HUMAN, EPPK1)	0/0/0	8/10/0	6.733
Caspase-14 (CASPE_HUMAN, CASP14)	0/0/0	7/11/0	6.683
Protein POF1B (POF1B_HUMAN, POF1B)	0/0/0	7/11/0	6.683
Malate dehydrogenase, mitochondrial (MDHM_HUMAN, MDH2)	0/0/0	7/7/4	6.633
Lon protease homolog, mitochondrial (LONM_HUMAN, LONP1)	0/0/0	8/7/3	6.612
DNA ligase 3 (DNLI3_HUMAN, LIG3)	0/0/0	6/9/3	6.512
tRNA selenocysteine 1-associated protein 1 (TSAP1_HUMAN, TRNAU1AP)	0/0/0	7/4/6	6.441
T-complex protein 1 subunit beta (TCPB_HUMAN, CCT2)	0/0/0	6/6/5	6.320
Ribosome biogenesis protein NSA2 homolog (NSA2_HUMAN, NSA2)	0/0/0	6/0/9	6.270
Dehydrogenase/reductase SDR family member 4 (DHRS4_HUMAN, DHRS4)	0/0/0	0/10/6	6.091
Serpin B3 (SPB3_HUMAN, SERPINB3)	0/0/0	8/8/0	6.067
Tripartite motif-containing protein 29 (TRI29_HUMAN, TRIM29)	0/0/0	8/8/0	6.067
RNA-binding protein PNO1 (PNO1_HUMAN, PNO1)	0/0/0	9/3/4	6.066
Probable ATP-dependent RNA helicase DDX47 (DDX47_HUMAN, DDX47)	0/0/0	3/6/7	5.978
60S ribosomal protein L28 (RL28_HUMAN, RPL28)	0/0/0	5/6/5	5.937
Zinc finger CCCH-type antiviral protein 1-like (ZCCHL_HUMAN, ZC3HAV1L)	0/0/0	7/0/7	5.845
2-Methoxy-6-polyprenyl-1,4-benzoquinol methylase, mitochondrial (COQ5_HUMAN, COQ5)	0/0/0	0/9/6	5.758
Peptidyl-prolyl cis-trans isomerase A (PPIA_HUMAN, PPIA)	0/0/0	6/9/0	5.633

^a^ The value was obtained from the mean normalized SC of the H1N1 N-terminus-truncated NS1 vector (tNS1) divided by that of the control vector (VC). Proteins with ratios larger than the mean plus two SDs (the ratios were above 5.5861) and detected in more than two replicates of the tNS1 group are defined as tNS1-interacting partners.

**Table 2 pathogens-11-00733-t002:** Enrichment analysis of biological processes for proteins interacting with the H1N1 N-terminus-truncated NS1 protein.

Biological Process ^a^	Identified Proteins Involved in Process	*p* Value
Cornification	FLG, DSP, CASP14, DSG1, PPL, EVPL, IVL	8.23 × 10^−8^
Keratinization	DSP, CASP14, DSG1, SFN, PPL, EVPL, IVL	2.44 × 10^−6^
Intermediate filament cytoskeleton organization	DSP, EPPK1, PPL, EVPL	9.07 × 10^−6^
Neutrophil degranulation	DSP, SERPINB3, CCT2, FLG2, CALML5, IMPDH2, DSG1, PPIA	3.68 × 10^−5^
Negative regulation of protein kinase activity	HSPB1, SFN, PPIA, YWHAG	3.35 × 10^−4^
rRNA processing	NSA2, DDX47, RPL12, RPL26, RPL28	5.33 × 10^−4^
Epidermis development	DSP, CASP14, CALML5, EVPL	5.56 × 10^−4^
Cytoplasmic translation	RPL12, RPL26, RPS11, RPL28	6.83 × 10^−4^
Peptide cross-linking	FLG, DSP, EVPL, IVL	9.07 × 10^−4^
Wound healing	DSP, EPPK1, PPL, EVPL	9.62 × 10^−4^

^a^ DAVID (2021 update) was applied to functionally annotate the proteins listed in Table 1 using the annotation category GOTERM_BP_DIRECT. Processes with *p* values ≤ 0.001 and false discovery rates ≤ 0.05 were considered significant.

**Table 3 pathogens-11-00733-t003:** Pathway analysis of the proteins interacting with the H1N1 N-terminus-truncated NS1 protein.

Reactome Pathway ^a^	Identified Proteins Involved in Pathway	*p* Value
Keratinization	FLG, DSP, CASP14, DSG1, PPL, EVPL, IVL	3.15 × 10^−5^
rRNA processing in the nucleolus and cytosol	DDX47, PNO1, RPL12, RPL26, RPS11, RPL28	1.81 × 10^−4^
Developmental biology	FLG, DSP, TUBA1C, CASP14, RPL12, DSG1, RPL26, RPS11, PPL, EVPL, RPL28, IVL	2.23 × 10^−4^
Neutrophil degranulation	DSP, SERPINB3, CCT2, FLG2, CALML5, IMPDH2, DSG1, PPIA	3.95 × 10^−4^
Programmed cell death	DSP, LMNA, DSG1, SFN, YWHAG	1.81 × 10^−3^
Eukaryotic translation termination/elongation	RPL12, RPL26, RPS11, RPL28	2.50 × 10^−3^
Rho GTPase signaling	DSP, CCT2, TUBA1C, DDX39B, STK38, DSG1, SFN, YWHAG	3.78 × 10^−3^

^a^ DAVID (2021 update) was applied to functionally annotate the proteins listed in Table 1. The knowledge base used was the Reactome Pathway Database. Pathways with *p* values ≤ 0.01 and false discovery rates ≤ 0.05 were considered significant.

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
