# Peer review of "Interactome Profiling of N-Terminus-Truncated NS1 Protein of Influenza A Virus Reveals Role of 14-3-3γ in Virus Replication"

_pathogens, 2022, doi:10.3390/pathogens11070733_

Round 1

Reviewer 1 Report

The manuscript presented by Kuo et al. describes the search, analysis, and identification of cellular proteins interacting with Influenza A NS1 protein. The method uses immunoprecipitation, which has been known for many years. A study of proteins binding to recombinant protein NS1 showed hundreds of candidates, many involved in critical processes in the host. The data is presented very well, but the manuscript does not offer a control where empty beads are subjected to the same procedure after IP. There is no WB band, but it does not mean that nothing binds to the beads. Considering the number of proteins identified, the experiment would answer the question of the validity of the targets.

Author Response

Dear Reviewer 1,

Thank you

Reviewer 2 Report

It has been previously reported that as a potent virulence factor, NS1 protein of influenza A virus inhibits type I interferon synthesis to overcome host defenses and to facilitate viral replication. In this manuscript, the authors demonstrate the interaction between 14-3-3γ and NS1 protein and a role of 14-3-3γ in influenza A virus replication. It is an interesting study because of the founding that 14-3-3γ may act in a different mechanism than other 14-3-3 members. But two vital points require attention and should be addressed as described below.

1.

From Figure 1A, both NS1 and tNS179-230 interact with 14-3-3γ, suggesting NS179-230 is required for the interaction. But in Figure 3, why PR8-NS1/1-98 mutant virus was used? It is reasonable that 14-3-3γ knockdown inhibits PR8 virus (including NS1 and tNS11-230) replication but not PR8-NS1/1-98 mutant virus replication since mutant virus has no the binding domain for 14-3-3γ. So, PR8-NS1/79-230 mutant virus should be used in Figure 3B. Also, PR8-NS1/79-230 mutant virus should be used in Figure 4 and 5. Lastly, NS179-230 plasmid should be used in Figure 6.

2.

For experiment condition in Figure 4, please detect IFN β mRNA and p-IRF3 at different points in time in order to cover the change mediated by 14-3-3γ knockdown.

Author Response

Dear Reviewer 2,

Thank you.

Round 2

Reviewer 1 Report

The corrected manuscript is much improved. The results are presented clearly and convincingly.

Reviewer 2 Report

Much appreciation for your efforts, very interesting research.